# Modeling Dynamics of Biological Systems with Deep Generative Neural Networks

## Abstract

Biological data often contains measurements of dynamic entities such as cells or organisms in various states of progression. However, biological systems are notoriously difficult to describe analytically due to their many interacting components, and in many cases, the technical challenge of taking longitudinal measurements. This leads to difficulties in studying the features of the dynamics, for examples the drivers of the transition. To address this problem, we present a deep neural network framework we call *Dynamics Modeling Network* or *DyMoN*. DyMoN is a neural network framework trained as a deep generative Markov model whose next state is a probability distribution based on the current state. DyMoN is well-suited to the idiosyncrasies of biological data, including noise, sparsity, and the lack of longitudinal measurements in many types of systems. Thus, DyMoN can be trained using probability distributions derived from the data in any way, such as trajectories derived via dimensionality reduction methods, and does not require longitudinal measurements. We show the advantage of learning deep models over shallow models such as Kalman filters and hidden Markov models that do not learn representations of the data, both in terms of learning embeddings of the data and also in terms training efficiency, accuracy and ability to multitask. We perform three case studies of applying DyMoN to different types of biological systems and extracting features of the dynamics in each case by examining the learned model.

## 1 Introduction

Many biological systems can be considered conceptually as stochastic dynamic processes. Examples include differentiation from stem cells to mature cellular lineages, responses to stimulation or drug perturbations, and responses to external stimulus or signaling. Such processes involve the slow and stochastic change in gene expression programs that alter over time, brain circuitry and neuronal communication where the firing of one set of neurons produces the firings of another set, or cancer progressions in which one subclone of cells out competes other clones. However, such biological systems are complex and in many cases, it is impossible to derive differential equations or analytical models for such processes in order to study them. In this paper, we propose to learn a generative neural network model of biological dynamic systems that we call a Dynamics Modeling Network, or DyMoN.

First, DyMoN provides a **representational** advantage by serving as an embodiment of the dynamics in lieu of a predetermined model, such as stochastic differential equations. This representation is deep and factored, in the sense that the "logic" of the dynamic transition is broken down into many increasingly abstract steps, each of which can be visualized or examined. In the cell trajectory context, a neural network can learn complex features that correspond to modules of genes that operate together to create a change in the expression program for the next cellular state. Further, the saliency of these features can be inferred from the Jacobian matrices of the neural network. Secondly, DyMoN provides a **generative** advantage as it generates new trajectories that have not been seen previously in the system. These can augment data and potentially reveal new paths (for instance novel differentiation paths in cellular development) that lead to desired outcomes. Third, it provides a **denoising** advantage, thanks to the the dimensionality reduction offered in neural networks, which can naturally denoise the high degree of noise in biological data, and in particular in biological trajectories. Fourth, the utilization of a deep network model offers a **multitasking** advantage: while previous models, such as HMMs, can simulate trajectories, and one may use PCA (or similar methods) to visualize trajectory

information, most previous methods are not designed or equipped to *simultaneously* learn multi-scale features (i.e., in many levels of abstraction), visualize the intrinsic underlying dynamics, and utilize them to generate new trajectories. Finally, the natural parallelizability of neural networks (e.g., with GPU-based implementations) offers **computational** advantages to deep learning approaches, as they can be used to process large volumes of noisy data as they become available for training. Furthermore, once trained, DyMoN can quickly generate new trajectories faster than most existing methods.

We have specifically designed DyMoN to learn a model of dynamics from the types of data available in biological systems: snapshot high dimensional data. By snapshot, we mean that biological data (for instance single cell RNA-sequencing data) is often collected only at one or a handful of time-points, thus the dynamics are either inferred from pseudotime (an axis of time progression based on different states of cell differentiation or other cellular programs) or from interpolation between discrete time points. Second, these types of trajectories are not a matter of sequence completion as is often done in a recurrent neural network, but rather the desired goal is to predict the near-future state of a cell given its current state. Thus, we propose a neural network framework which learns fixed-memory stochastic dynamics (e.g., memoryless or $n$-th order Markov process) in an observed system, and train this network by providing a current state (or $n$ states) along with a probability distribution of next states. We penalize the network using maximal mean discrepancy (MMD) as a probabilistic distance between the desired and generated outputs. Within this framework, we show three variations of architectures that can realize this framework within three vastly different biological systems.

We train the DyMoN on three biological systems: dynamic calcium imaging data of neurons from a mouse visual cortex, mass cytometry of developing T cells in the mouse thymus, and single-cell RNA sequencing of human embryonic stem cells developing in an embryoid body. We interrogate the weights of the model as a "transparent box" to understand the dynamics of the system learned by the network in order to both verify known gene-gene interactions in T cells and to uncover novel network interactions in neurons. We further show that the DyMoN is able to generate trajectories in biological data originally measured in discrete snapshots, and use this generative capacity to develop novel hypotheses for cellular programming in embryonic stem cell differentiation.

## 2 DYNAMICS MODELING NETWORK (DYMON)

Let $\mathcal{X} \subseteq \mathbb{R}^d$ be a finite dataset of $d$-dimensional states, and let $\mathcal{T} \subseteq \mathcal{X} \times \mathcal{X}$ be a (finite) collection of transitions between these states. Such transitions can either be observed by sampling a dynamical system, or constructed by geometry-revealing diffusion methods such as diffusion maps (Coifman & Lafon, 2006), Laplacian eigenmaps (Belkin & Niyogi, 2002), diffusion geometry methods (e.g., Wolf et al., 2012; Wolf & Averbuch, 2013; Coifman & Hirn, 2014; Talmon & Coifman, 2015), and PHATE (Moon et al., 2017). We propose a deep learning approach for learning the dynamics in $\mathcal{T}$ and the geometry represented by them as a stochastic velocity vector field of a Markov process. To do this, we use a feed-forward neural network we call Dynamics Modeling Network (DyMoN).

DyMoN is formed by a cascade of linear operations and nonlinear activations. These are controlled by optimized network weights, which we collectively denote by $\theta \in \Theta$, where $\Theta$ represents the space of possible weight values determined by the fixed network architecture. To capture the dynamics in $\mathcal{T}$, DyMoN learns a *velocity vector* $\Delta_\theta(x) \in \mathbb{R}^d$ and uses it to define a transition function $T : \mathbb{R}^d \times \Theta \to \mathbb{R}^d$ that generates a Markov process

$$x_t = T(x_{t-1}, \theta) = x_{t-1} + \Delta_\theta(x_{t-1}) \qquad t = 1, 2, 3, \ldots,$$

given an initial state $x_0 \in \mathcal{X}$ and optimized weights $\theta \in \Theta$. Further, to introduce stochasticity, we treat the output $\Delta_\theta(x_{t-1})$ as a random vector whose probability distribution determines the conditional probabilities $P(x_t|x_{t-1})$ of the Markov process. We provide three possible implementations of this framework in Figure 10.

Given a fixed architecture, we now describe how DyMoN is trained using the data points in $\mathcal{X}$. Consider a source state $x \in \mathcal{X}$ with multiple transitions from it leading to target states $Y_x = \{y : (x, y) \in \mathcal{T}\}$. Let $\mathcal{P}_x(y) = P(y \mid x)$, whose support is $Y_x$, be the conditional probability of transitioning from $x$ to $y$. For ease of notation, we allow repetitions in the set notation of $Y_x$, and assume that data points in $Y_x$ are indeed distributed according to $\mathcal{P}_x(y)$ and are sufficient for estimating $\mathcal{P}_x(y)$. Further, we assume that these transitions are smooth in the sense that if $x$ is similar to $x'$ then $P(y \mid x)$ is similar to $P(y \mid x')$. Therefore in practice we may replace $Y_x$ in the following

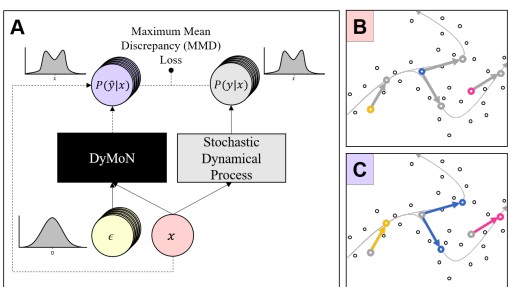

Figure 1: (**A**) Schema of DyMoN architecture where $x$ represents the neural network input vector, $\hat{y}$ represents the predicted network output in (1), and $\epsilon$ is a random Gaussian noise vector. $P(y|x)$ represents the distribution of outputs from many iterations of the stochastic dynamical process given $x$, and $P(\hat{y}|x)$ represents the distribution of outputs of DyMoN given $x$ and many different noise vectors. (**B**) Example input states. (**C**) Learned transition vectors (arrows) and output states from DyMoN.

training procedure with $\cup_{x' \approx x} Y_{x'}$, which includes target points from neighbors of $x$ to increase the robustness of DyMoN.

The stochastic output of DyMoN is enabled by a random input vector $\varepsilon \in \mathbb{R}^n$ sampled from a simple (e.g., normal or uniform) distribution $\mathcal{F}$ with zero mean and unit variance. This input can be explicitly written into DyMoN function as

$$T_\varepsilon(x, \theta) = x + f_\theta(x, \varepsilon), \tag{1}$$

where $f_\theta : \mathbb{R}^d \times \mathbb{R}^n \to \mathbb{R}^d$ represents suitable feed forward layers for combining the training input $x$ with the random input $\varepsilon$ to provide an instantiation of the transition velocity vector $\Delta_\theta(x)$.

Given $x \in \mathcal{X}$ and $\theta \in \Theta$, one can consider the distribution $\widehat{\mathcal{P}}_x^{(\theta)}$ of the random variable $T_\varepsilon(x, \theta)$, $\varepsilon \sim \mathcal{F}$ and estimate it by $m$ i.i.d. instantiations $\boldsymbol{\epsilon} = \{\varepsilon_j \sim \mathcal{F}\}_{j=1}^m$ passed through the network to form $\hat{Y}_{(x,\theta)}^{(\boldsymbol{\epsilon})} = \{T_{\varepsilon_1}(x, \theta), \ldots, T_{\varepsilon_m}(x, \theta)\}$. To conform with the training data, DyMoN is optimized so that this distribution approximates the distribution $\mathcal{P}_x$, as captured from the training data by $Y_x$. This optimization is given by $\arg\min_\theta \mathbb{E}\left[\{H_x(\theta) : x \in \mathcal{X}\right]$ with $H_x(\theta) = \text{MMD}(\widehat{\mathcal{P}}_x^{(\theta)}, \mathcal{P}_x)$ (see supplemental material for details on MMD). The MMD is computed using $\hat{Y}_{(x,\theta)}^{(\boldsymbol{\epsilon})}$ and $Y_x$. Given trained weights $\theta \in \Theta$ and an initial state $x_0 \in \mathcal{X}$, a random walk is generated by $x_t = T_{\varepsilon_t}(x_{t-1}, \theta)$, where $t = 1, 2, \ldots$, and $\varepsilon_1, \varepsilon_2, \ldots \overset{\text{i.i.d.}}{\sim} \mathcal{F}$.

We note that DyMoN can be used to learn deterministic time series by considering Dirac conditional probabilities in the Markov process, essentially ignoring the random input. Additionally, DyMoN can be extended to encode high-order Markov processes, which have fixed-size memory rather than being memoryless. To this end, an $n$-th order DyMoN takes $n$ input states and have the distribution of $\Delta_\theta(x_{t-1}, x_{t-2}, \ldots, x_{t-n})$ determine the conditional probability $P(x_t|x_{t-1}, x_{t-2}, \ldots, x_{t-n})$ of an $n$-th order Markov process. The training and application of both a deterministic and a high-order DyMoN extend naturally from the stochastic memoryless (i.e., first-order) DyMoN.

# 3 BIOLOGICAL CASE STUDIES

## 3.1 BIOLOGICAL CASE STUDY 1: CALCIUM IMAGING OF VISUAL CORTEX NEURONS

We apply DyMoN to dynamic calcium imaging data from 33 Somatostatin-expressing GABAergic (SOM) interneurons from a mouse visual cortex  (Urban-Ciecko & Barth, 2016). SOM interneurons are a unique set of GABAergic cells and represent about 25% of all cortical interneurons. and are characterized by high basal firing activity. They are also thought to be connected in dense networks, though the architecture of these networks is not well-known. The data consists of a total of 129,500 frames at 30Hz, lasting 72 minutes, each containing a recording of the change of fluorescence (dF/F) indicating calcium activity over time . We train DyMoN to predict the the next state (which we define as 10 frames later) given the current state of the 33 neurons.

Surprisingly, though SOM cells are not connected to each other by chemical or electrical synapses and do not receive strong feedforward input, we see co-activation relationships in the data. We see in Figure 2A that neurons 9 and 18 are strongly affected by two large clusters of neurons in opposite directions. There is a group of nine neurons which activate 9 and 18, with neuron 19 being the strongest activator. A second group of 9 neurons which inhibit neurons 9 and 18, with neuron 4 being

the strongest inhibitor. We speculate that these interactions are the result of either 1. co-activation by lateral excitatory inputs from superficial pyramidal neurons, 2. co-modulation by neuromodulators, or 3. top-down feedback input regulating the activity of a specific subset of SOM cells (Yavorska & Wehr, 2016).

A subset of the SOM cells appear to function as hubs in an excitatory pattern (see Figure 2B). Early in postnatal development, a subset of inhibitory neurons function as hub neurons and regulate the activity of surrounding neurons. Hub neurons have mainly been characterized in the hippocampus and entorhinal cortex, and a hub role for sensory cortical interneurons would be highly novel (Bonifazi et al., 2009). Our future work involves examining whether these results hold in additional mouse samples.

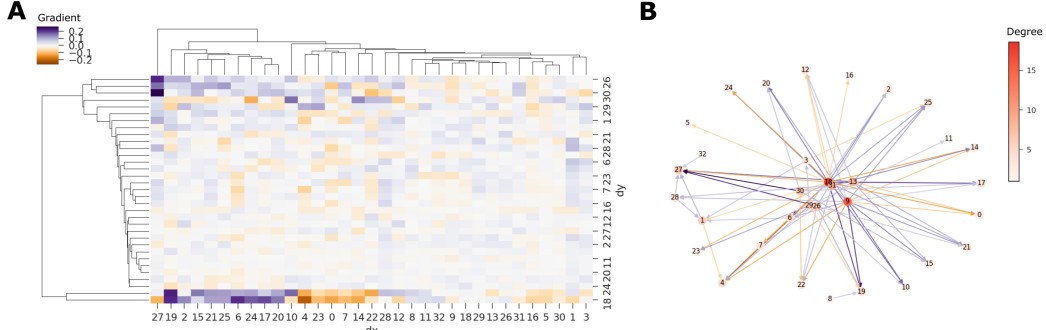

Figure 2: Jacobian of DyMoN's transition vector with respect to network inputs on mouse neuronal activations. (**A**) Biclustered heatmap of gradient of the transition vector with respect to the network inputs, with the transition vector on the vertical axis and the inputs on the horizontal axis. (**B**) Graph built on thresholded gradients (gradients with absolute value less than 0.04 were excluded). Nodes are colored by their degree.

## 3.2 BIOLOGICAL CASE STUDY 2: T CELL DEVELOPMENT IN THE THYMUS

Next, we use DyMoN to learn transitions in single-cell data. Single-cell data has recently gained popularity in biology as a way of dissecting cellular heterogeneity. Recent works such as Diffusion Pseudotime (Haghverdi et al., 2016), Wanderlust (Bendall et al., 2014), and Wishbone (Setty et al., 2016) take the view that cells from a single experimental sample are at different stages of development and derive a pseudo-temporal developmental ordering to the cells. The idea of pseudo-time is that while we cannot follow a single cell through time with current high-dimensional single cell technologies, the entire population of cells can be used to derive potential cell trajectories within a sample.

Here, we use a mass cytometry dataset measuring actively developing T cells, which are adaptive immune cells, obtained from a mouse thymus (Setty et al., 2016). We preprocess this data using the MAGIC algorithm (van Dijk et al., 2018). After preprocessing, we are left with 17,000 cells, measured in 33 protein dimensions (Figure 3). We then train DyMoN on a trajectory obtained by using a diffusion-based Markov affinity matrix (as described in Moon et al. (2017)) and the ordering obtained by Wishbone.

After training we sample trajectories by initializing DyMoN with undifferentiated cells. We obtain two types of trajectories. To investigate the difference between the two trajectories we examine their marker expression as a function of trajectory progression (Figure 3ii, iii). Both trajectories start out with low CD4 and CD8, which is expected form naive T cells. Both trajectories then increase in CD4 and CD8, signaling the change to the double-positive T cells. Then, at this point, we find that the two trajectories diverge with the main difference between the two trajectories that in one CD8 goes down whereas in the other CD4 goes down. This signifies the transitions into T helper cells (CD4+/CD8-) or cytotoxic T cells (CD4-/CD8+) respectively.

To investigate the internal representation that is learned by DyMoN, we compute the Jacobian of the transition vector with respect to the inputs, at two different points on the trajectory: at the branch point (Figure 3Di) and at the CD8+/CD4- branch (Figure 3Dii). DyMoN is capable of learning

different associations between genes at different points in the ambient space. We find a negative association between CD4 and CD8 in the Jacobian obtained at the branch point, reflecting the decision between downregulating either CD4 or CD8. In addition, we find that in the Jacobian of the CD8+ branch Gata3, a developmental marker, goes down with CD4 and CD8 confirming the developmental characteristic of the trajectory. Finally, in the same CD8+ branch Jacobian, Foxp3 and CD25 are positively associated, which is consistent with their role as regulatory T cell markers. The concordance of our analysis with the established literature on T cell development provides validation that DyMoN can both learn meaningful trajectories, and can be further interrogated to provide insights into the network's dynamical model of the system.

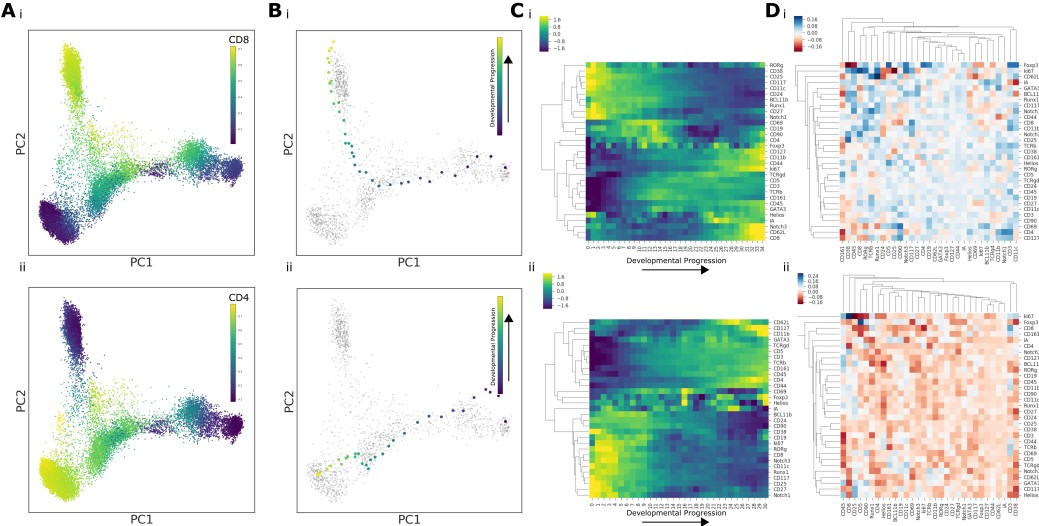

Figure 3: DyMoN on mass cytometry data of T cell development in the thymus. (**A**) PCA plots of all 17,000 cells colored by CD8 (**Ai**) and CD4 (**Aii**) expression. (**B**) PCA plots with all cells in grey and DyMoN trajectories in color. **Bi** shows DyMoN trajectory of CD4+/CD8- T helper cells, and **Bii** shows DyMoN trajectory of CD4-/CD8+ cytotoxic T cells. (**C**) Shows row z-scored heatmaps of marker expression as a function of the trajectory for each of the two DyMoN trajectories with hierarchically clustered genes on the rows and cells on the columns. (**D**) Heatmaps of the Jacobians obtained at the branch point (**Di**) and at the end of the CD8+/CD4- branch (**Dii**).

## 3.3 BIOLOGICAL CASE STUDY 3: HUMAN EMBRYONIC STEM CELL DIFFERENTIATION

In order to apply DyMoN's generative capabilities on a biologically novel system, we generated single-cell RNA sequencing data of human embryonic stem cells (hESCs) differentiating in an embryoid body (EB) system. EB differentiation has been shown to mirror early human development in numerous cell types (e.g. neuronal, hematopoietic, muscle, pancreatic cells, etc), and is used as a model to develop cellular reprogramming protocols (Bibel et al., 2007; Nakano et al., 1996; Kania et al., 2004; Geijsen et al., 2004).

To investigate potentially novel reprogramming strategies in the EB system with DyMoN, we measured approximately 31,000 cells, using the 10x Chromium platform, equally distributed over a 27-day differentiation time course, and preprocessed the cells with MAGIC (van Dijk et al., 2018). Because this dataset has numerous lineages, it is not amenable to training by diffusion pseudotime. Instead, we train the DyMoN on Markovian transitions based on the diffusion geometry of the dataset. We sample neighbors $\mathbf{y}$ of $x$ over weighted affinities defined by a Gaussian kernel, retaining only those neighbors for which $y$ is defined to be "later" than $x$, where time is defined here as a smoothed estimate of the discrete time variable denoting development within the experimental time course.

In Figure 4, we show two representative trajectories generated through the data by DyMoN, one of which samples neural progenitor development (A), and the other bone progenitor development (B), both starting out as embryonic stem cells. We show in a heatmap of genes with high mutual information with the path position that each of these paths initially express known stem cell markers

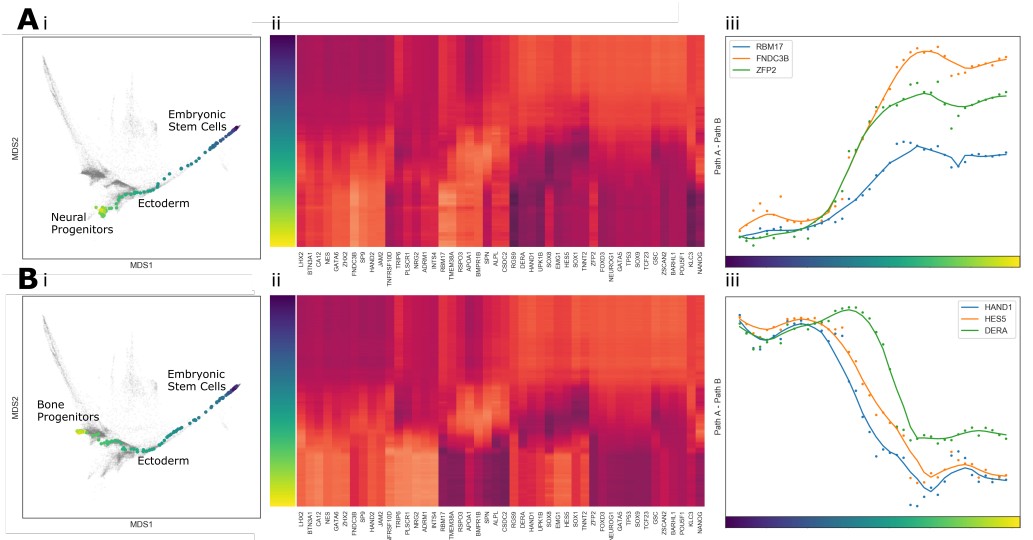

Figure 4: Trajectories generated by DyMoN sampling the neural progenitor (A) and bone progenitor (B) cell states. Trajectories are shown both as MDS against the training data (i) and as heatmaps of selected transcription factors (ii). From these trajectories, we propose a novel cellular programming protocol shown in (iii).

NANOG and POU5F1, and follow a common differentiation transition into the ectoderm (LHX2). Following this, the two paths diverge, with path A taking the neural progenitor branch distinguished by SOX1, and path B taking the bone progenitor branch distinguished by ALPL. We observe a novel set of transcription factors distinguishing each stage of differentiation in these generated trajectories (e.g. RBM17, FNDC3B, HAND1, and HES5, see Figure 4iii), and propose these transcription factors for a potentially novel reprogramming protocol to obtain each respective mature cell type. In this system, DyMoN provides a novel form of hypothesis generation enabled by the deep abstract model of the differentiation process, enabling researchers to perform more informed validation experiments based on hypotheses generated in an unsupervised and data-driven manner.

## 4 EMPIRICAL VALIDATION

In this section, we demonstrate the performance and accuracy of DyMoN on a variety of datasets including simulated datasets and the Frey faces dataset (Roweis & Saul, 2000). We show that DyMoN is able to learn processes that are harmonic or chaotic, ergodic or null-recurrent, and compare its performance to a range of other methods used to perform prediction in dynamical systems. We further show that DyMoN's output and hidden layers can be interrogated to learn about the process being modeled. Specifically, we show that we can reliably use DyMoN to 1. learn the transition probabilities at each state; 2. generate stochastic trajectories within a system; 3. visualize the data; and 4. learn the relevant features that drive the transitions. For training details, see the supplementary material.

**Learning transition probabilities:** We first demonstrate DyMoN on two test cases: a circle, which is traversed directionally and non-directionally; and a tree, which is traversed directionally starting from the root. The tree is generated by creating points sequentially along branches at random angles with one another, and then adding Gaussian noise. In both cases, the inputs to the network are two-dimensional Euclidean coordinates. Training examples are drawn from the tree where transitions crossing branch points select a branch with probability 0.5.

Figure 5 shows vector fields for these two examples indicating the distribution of transitions at each state and at unobserved states as predicted by DyMoN. DyMoN is able to randomly select a direction of travel at each branch point, allowing it to walk along a data manifold in the correct direction. Empirical testing on these test systems also motivates our choice to include a skip connection from

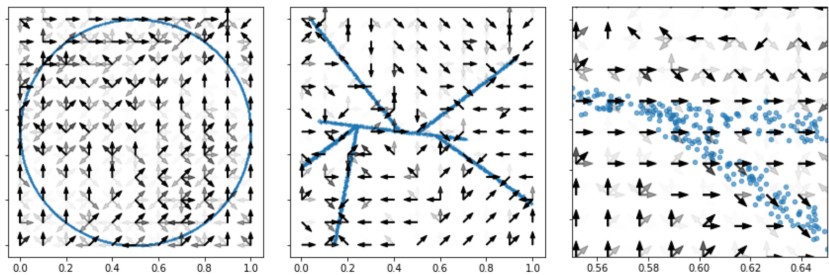

Figure 5: Vector fields indicating the predicted transitional direction by DyMoN on circle (left) and tree (center) datasets. The right-hand branch point of the tree is shown at higher resolution in the right panel. We run DyMoN at discrete points in the state space, with arrows pointing in the direction of prediction. We run DyMoN 40 times and show darker arrows in the more frequently predicted directions. DyMoN is trained to move both clockwise and counter-clockwise on the circle. For the tree, it begins at the center-left and travels down each of the branches.

the inputs to the outputs, allowing the network to learn the velocity of the transition function rather than the position transition itself (shown in the supplemental material).

**Trajectory generation:** We now demonstrate the ability of an $n$th order DyMoN to generate paths on a single and double pendulum. Since these processes are deterministic, we do not add noise to the input of DyMoN and use the mean squared error as the loss function. For the single pendulum, we use a second-order DyMoN with the current and previous angle of the pendulum as inputs. For the double pendulum, we use a third-order DyMoN with the current and last two angles of both pendulums as inputs.

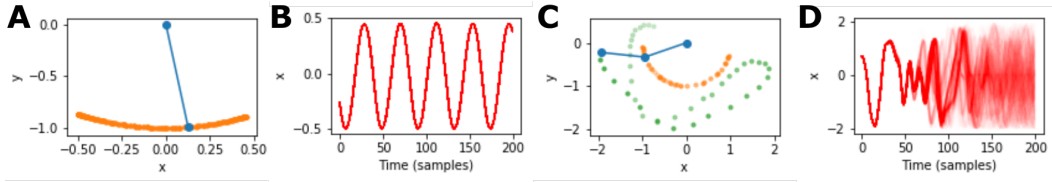

Figure 6: Paths generated by the second-order DyMoN trained on a single and double pendulum. A single trajectory is shown for each system in (**A**) and (**C**) respectively. In (**B**) and (**D**) we show the x coordinate of 500 generated paths starting from an epsilon-difference for the single and the lower of the double pendulums, respectively.

Figure 6 shows the Euclidean coordinates of DyMoN-generated paths of both pendulums over time, with only the second pendulum shown in the case of the double pendulum. Both predicted pendulums show smooth trajectories, with the single pendulum showing periodic behavior and the double pendulum showing chaotic behavior.

We also trained DyMoN on the Frey faces dataset (Roweis & Saul, 2000) in order to demonstrate DyMoN's capacity to learn an empirical model of a stochastic system for which no generating distribution exists. Figure 7 shows 1000 samples generated by DyMoN on the Frey faces dataset visualized using PCA. The first 1000 samples are discarded. Ten generated samples with uniform spacing in time are also shown. DyMoN samples a large range of states and generates a realistic new trajectory.

**Stationary distribution comparison to other methods:** We train the DyMoN to sample a Gaussian mixture model in order to show that DyMoN reliably learns to generate samples of the system which match the expected data distribution, thus avoiding the "mode collapse" problem which affects Generative Adversarial Networks. We generate training data from a 1-dimensional Gaussian mixture model using Metropolis-Hastings sampling, and compare the performance of Recurrent Neural Networks (RNNs), Hidden Markov Models (HMMs) and Kalman Filters (KFs) on this simple stochastic system. Figure 8 shows 50,000 points generated in a chain from each method, and we compare the Earth Mover's Distance (EMD) of these chains to a distribution of 200,000 points

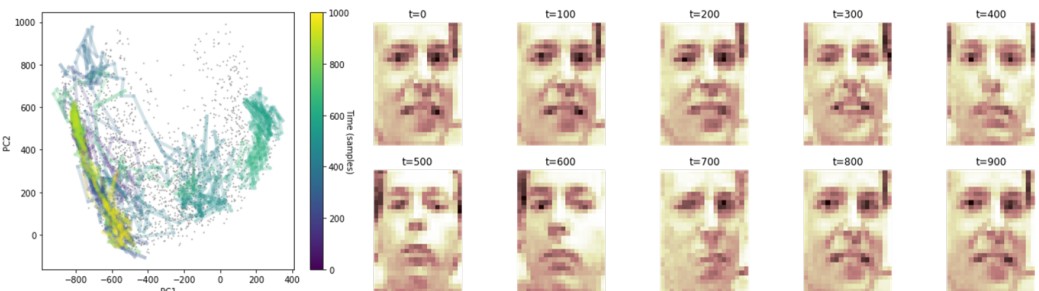

Figure 7: Chain generated by the DyMoN visualized on a PCA embedding of the Frey faces dataset (left). The chain is shown in color (indicating time) superimposed over the training data in gray. 10 equally spaced faces generated by DyMoN are also shown (right).

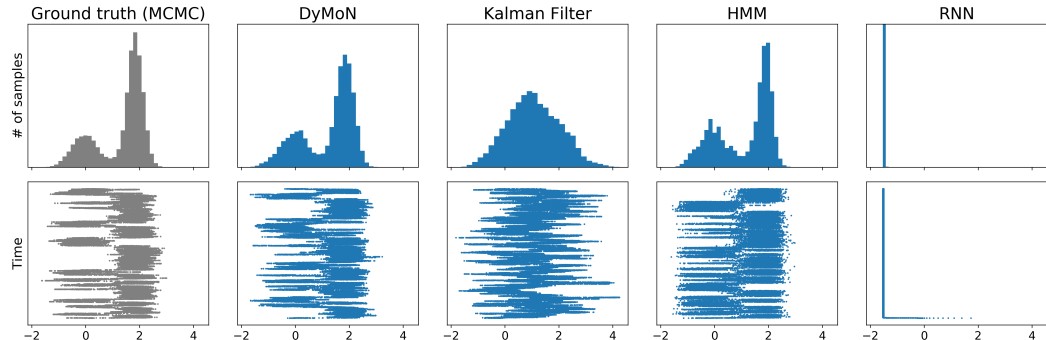

Figure 8: Chain of samples (top) and marginal distribution (bottom) drawn from Markov-Chain Monte Carlo (MCMC), DyMoN, Recurrent Neural Network (RNN), Hidden Markov Model (HMM) and Kalman Filter (KF) when trained on the Gaussian Mixture Model.

generated by MCMC in Table 5. DyMoN provides performance on par with the HMM and MCMC, both in terms of accuracy of sampling the distribution (EMD) and inference time. RNNs fail to capture the stochasticity of the system, falling into an infinite loop reproducing the same data point. Kalman filters are inherently difficult to train on a system without predefined states and transitions, and as such under-perform in an unsupervised setting. Additionally, although the marginal distribution is similar, the dynamics of the HMM fail to capture the full variability of the MCMC sampling.

**Pointwise sample comparison to other methods:** We train the a DyMoN with 2 convolutional layers with 5x5 filters and 2x2 max pooling, a single fully connected layer of 3 hidden nodes, and 2 deconvolutional layers on a video of a rotating teapot (Weinberger et al., 2004). We compare the performance of RNNs, HMMs and KFs to DyMoN on this higher-dimensional example. We produce a single transition from each frame in the time series (giving a chain of prior states to the HMM, RNN and KF in order to facilitate their estimates of the current state) and measure the mean squared error between the produced output and the true frame, ten time points after the current state. Due to the relatively little training data available (400 samples), both classical statistical learning methods (HMM, KF) have difficulty generating accurate samples from this system given a single snapshot. RNNs perform well on the large majority of frames when given a sequence as input, but fail entirely when given only a single frame as input on inference, making them inherently unsuitable to inference in snapshot biological systems where sequential input is not available. Additionally, DyMoN is able to generate images faster than all other methods, and is only slower in training than the HMM, which produces significantly lower quality images. Examples of images produced by each method are shown in Figure 11.

**Visualization:** A DyMoN with a low-dimensional latent layer can also be used to produce a visualizable embedding of the data. Figure 9 shows the embedding layer of the DyMoN trained on the teapot data produces a single trajectory homeomorphic to a circle, while the PCA embedding of the same dataset produces spurious intersections and branches in the data. We see that the RNN trained

| | MSE | TRAINING (CPU S) | INFERENCE (CPU S / FORWARD PASS) |
|---|---|---|---|
| DYMON | $1.8 \pm 0.01$ | 8 MIN | 1.2 S |
| RNN (SEQUENCE) | $2.3 \pm 0.20$ | 8.7 H | 3.0 S |
| RNN (SNAPSHOT) | $9.1 \pm 0.04$ | — | 2.9 S |
| KF | $10.2 \pm 0.07$ | 300 H | 175 S |
| HMM | $5.4 \pm 0.06$ | 28 S | 192 S |

Table 1: Performance of various learning methods for dynamical systems on generating frames from the teapot dataset.

with a similar architecture (a 64 node LSTM layer is added to both sides of the 3 node layer) does not contain an interpretable embedding, as the RNN does not simply model the transition and has memory that may confound the embedding.

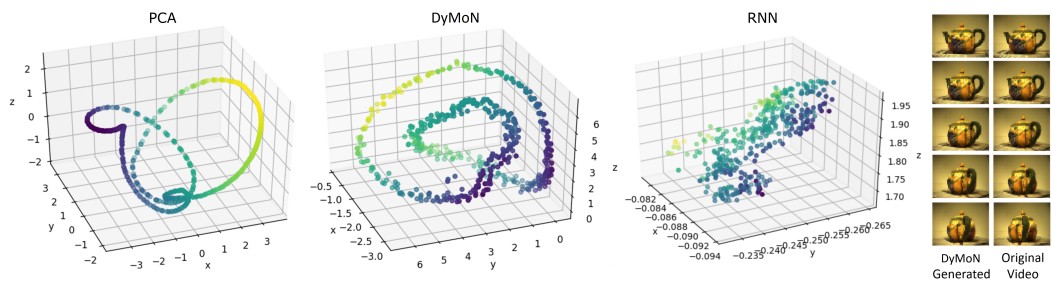

Figure 9: Embedding of the teapot video dataset colored by the x axis using PCA (left), a three-node hidden layer of DyMoN (center), and a three-node hidden layer of an RNN (right). We also show sequential images from the original video and a DyMoN-generated chain (far right).

## 5 CONCLUSION

Here we presented a framework, which we call DyMoN, for designing neural networks that can model stochastic dynamics from biological data. The DyMoN framework is well-suited for creating a deep model of $n$-th order Markovian stochastic dynamics from the types of data that occur in biological settings, especially in systems for which the generative process cannot be described by differential equations. In addition to actual longitudinal samples of dynamics, this can include noisy or sparse data, snapshot data where neighborhoods represent probabilistic next states, pseudotime trajectories created over data, or even trained Markov models. The flexibility of this framework is enabled by several aspects of DyMoN. First, since the networks encode Markovian dynamics, they do not require long histories that are difficult to obtain in biological data. Secondly, the MMD penalty used to train the next-state generation, can enforced via samples or a known probability distribution, making the networks trainable on top of other more shallow models such as diffusion operators and pseudotime inferences. We show that creating a deep model of biological stochastic dynamics enables several different analyses of such systems, including the generation of new trajectories to augment the data, extraction of feature dependencies of the dynamics, and visualization of the dynamic process, as well as denoising trajectories from naturally noisy data. We believe that DyMoN will enable inference of driving forces (genes, transcription factors, mutations, etc) of progression dynamics from the large amounts of biomedical data now being generated in many settings.

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

## A  BACKGROUND

**Markov Processes:** Markov processes are stochastic memoryless models that describe dynamic systems, i.e., each state depends only on the last state. However, higher order processes can be modeled by making the state dependent on several previous states. A deterministic Markov chain maps each state to only one next state whereas a stochastic Markov chain models a probabilistic transition such that at each state multiple transitions are possible. DyMoN is a neural network trained to replicate a stochastic (with input noise) or deterministic (no input noise) Markov process.

**Maximum Mean Discrepancy:** Divergences, such as KL divergence, are used to measure distances between probability distributions. However, they require density estimation and as such divergences are hard to compute for high dimensional systems. Maximum Mean Discrepancy (MMD) (Gretton et al., 2012) offers a solution by using the kernel trick to circumvent the curse of high dimensionality. Instead of comparing empirical distributions, MMD is defined on the inter- and intra-sample pairwise affinities, which are computed using a kernel. With MMD we can therefore compute the distributional distance (i.e. divergence) between two samples, without the need for density estimation. This distance is defined as

$$MMD^2(\mu, \nu) = \iint k(x, x')d\mu d\mu + \iint k(y, y')d\nu d\nu - 2\iint k(x, y)d\mu d\nu$$

where $k(\cdot, \cdot)$ is a kernel function and $x$ and $y$ are sampled from the two distributions represented by the probability measures $\mu$ and $\nu$. In practice, this distance is estimated using summation over two finite sets of samples.

MMD has been used in Dziugaite et al. (2015) to translate samples from one distribution into another distribution using a deep neural network architecture. Samples were generated by sampling random values from some simple distribution, e.g. Gaussian, and running them through a deep neural network. The network was trained by minimizing the MMD between generated samples and real samples. In Shaham et al. (2017) this architecture was extended by using a residual network with the application of removing undesirable batch effects that are associated with measurements. We use MMD to learn a conditional distribution that represents the possible next states in the Markov chain.

**Calcium Imaging Data (Urban-Ciecko & Barth, 2016) Details:** To express GCaMP in SOM+ cells, the mouse was anesthetized with 1- 2% isoflurane mixed with pure oxygen, and we injected the two 100nL injections of adenoassociated virus AAVdj-EF1a-fDIO-GCaMP6m in primary visual cortex (V1) at a rate of 75nL/min. For implantation of the imaging window, the mouse was anesthetized using a mixture of ketamine (100 mg/kg) and xylazine (10 mg/kg), and a 3 mm diameter craniotomy was opened over V1, where we inserted and fixed a small rectangular glass piece attached to a 5mm circular cover glass into the craniotomy. A custom titanium head post was secured to the skull with Metabond. Two weeks after implantation, mice were placed on the wheel and head-fixed under the microscope objective. Imaging was performed using a resonant scanner-based two-photon microscope (MOM, Sutter Instruments) coupled to a Ti:Sapphire laser (MaiTai DeepSee, Spectra Physics) tuned to 920 nm for GCaMP6. Images were acquired using ScanImage at  30 Hz, 512x512 pixels (580x580 um). Imaging of layer 2/3 was performed at  150-300 um depth relative to the brain surface. Mice ran freely on the wheel, and after 10 mins of recording, were presented with visual stimuli (drifting gratings with vary contrasts) for 50 mins (40 presentations of each contrast, contrasts ranging from 0%, i.e. gray screen, to 100%, steps of 10%). After the stimulation session was done, we recorded for another 10 mins of spontaneous activity. Analysis of imaging data was performed using ImageJ and custom routines in MATLAB. Motion artifacts and drifts in the Ca 2+ signal were corrected with the moco plug-in in ImageJ (Dubbs et al., 2016), and regions of interest (ROIs) were selected as described in (Chen et al., 2013). All pixels in a given ROI were averaged as a measure of fluorescence, and the neuropil signal was subtracted.

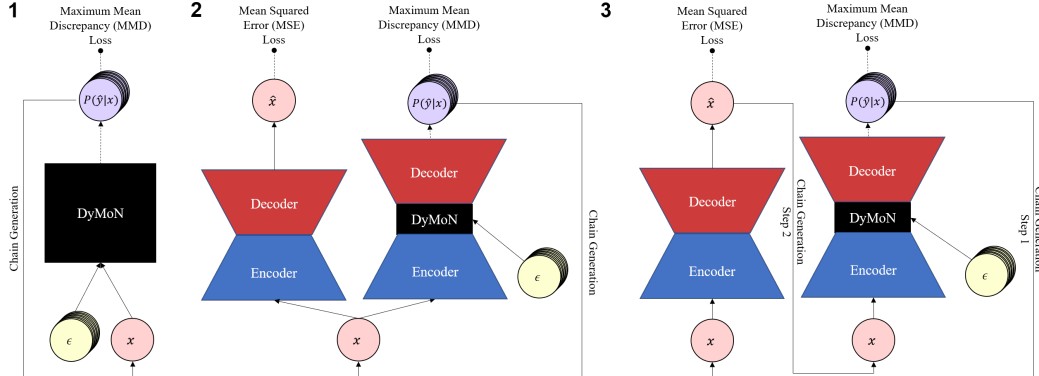

Figure 10: Alternative DyMoN architectures. The stochastic output can be generated in the ambient space (1) or the latent space of an autoencoder (2, 3). Chains of samples are generated by feeding DyMoN output back in as a new input; this can be passed additionally through the encoder and decoder as a denoising step (3).

## B  RELATED WORK

**Recurrent Networks (RNNs):** RNNs are trained to predict the next state of a sequence and have been used for text analysis (Fernández et al., 2007), speech recognition (Amodei et al., 2016), and language modeling (Józefowicz et al., 2016), as well as other tasks that operate on time series data. In contrast to DyMoN, RNNs typically require a sequence or history to predict the next state.

**Stochastic Generative Networks:** There have been other networks that use stochasticity to learn conditional distributions. For example, given an input $X$ and a stochastic input to a middle layer, a variational autoencoder (VAE) (Kingma & Welling, 2014) learns to transform these inputs into a Gaussian centered at a maximum likelihood point from which $X$ is derived. Thus the variational network can denoise samples and also generate samples "like" given samples. However, VAEs focus on generating data points instead of transitions. Several stochastic RNNs have also been proposed (Bowman et al., 2016; Chung et al., 2015). For example, in Goyal et al. (2017), a stochastic generative RNN is trained to learn a process that converges to the full data distribution within a small number of steps given a simple initialization. In contrast, DyMoN learns the transitional probabilities from state to state of a Markov process instead of its stationary distribution.

**Time Delay Neural Networks:** DyMoN also has some conceptual connection to Time Delay Neural Networks (TDNNs) which include a contextual window samples as input. However, the main focus of TDNNs are to classify patterns with shift-invariance, such as recognizing phenomes in speech (Waibel et al., 1989). TDNNs generally achieve this by taking a time convolution through windows of time to train a classifier. In contrast, we focus on learning Markov processes with the goal of generating plausible next-states and analyzing the dynamics that drive these transitions.

## C  DYMON DETAILS

**DyMoN architecture:** One could envision many neural network architectures which fit the schema of the DyMoN that we propose. In this paper, we use three alternative models; these are shown in Figure 10. The first architecture generates transitions on the ambient space, most suitable to data of low dimensionality. The other two architectures generate transitions on the latent space of an autoencoder, which learns a latent space which, due to gradient descent propagating through the encoder and decoder, learns a latent space more suited to generating transitions than either the ambient space or latent spaces learned by other dimensionality reduction methods. Additionally, the generation of points in a sampled trajectory can be passed through the encoder and decoder in order to prevent accumulation of error over the course of many samples.

**Training the DyMoN:** We train DyMoN using leaky ReLU activations on the hidden nodes and linear activations on the residual output nodes. Stochastic DyMoNs are trained with Gaussian noise inputs, and all DyMoNs are trained with Gaussian corruption noise. To compute the MMD loss, we

| DATA SET | ARCHITECTURE | HIDDEN LAYERS | STEP SIZE |
|---|---|---|---|
| CIRCLE | 1 | 1x6 | $0 \pm 20$ |
| PENDULUM | 1 | [8, 16, 8] | 1 |
| TEAPOT | 2 | 2xCONV, 1x3, 2xDECONV | 10 |
| MIXTURE MODEL | 1 | 3x64 | 1 |
| TREE | 1 | [64, 128, 64] | $100 \pm 20$ |
| FREY FACES | 1 | [512, 1024, 512] | $0 \pm 12$ |
| DOUBLE PENDULUM | 1 | [64, 128, 64] | 1 |
| CALCIUM IMAGING | 1 | 3x128 | 10 |
| T CELL CYTOF | 2 | 2x256, 3x256, 2x256 | $80 \pm 20$ |
| hESC scRNA-SEQ | 3 | 2x128, 3x128, 2x128 | DIFFUSION |

Table 2: DyMoN architecture and training details. Architecture refers to alternative DyMoN architectures shown in Figure 10. Where step size is given, samples are provided to DyMoN as $(x_t, x_{t+step\_size})$.

| DATA SET | EPOCHS | WITH | WITHOUT |
|---|---|---|---|
| CIRCLE | 2200 | 1.3E-8 | 4.7E-5 |
| PENDULUM | 750 | 1.9E-6 | 3.9E-5 |

Table 3: Training loss for DyMoN with and without skip connection.

use a multi-scale Gaussian kernel (Bousmalis et al., 2016) with 19 bandwidths ranging from $1e-6$ to $1e6$, evenly spaced on a log scale. The kernel is computed separately for each bandwidth and then MMD is computed on the sum of the kernels. The Adam optimizer (Kingma & Ba, 2015) is used for stochastic gradient descent in all cases.

**Empirical validation of the skip connection:** We trained a DyMoN on the circle and pendulum datasets both with and without a skip connection. The resulting training losses are shown in Table 3. Including a skip connection reduces the training loss.

**Training time:** Table 4 shows DyMoN training times for the datasets considered.

| DATA SET | EPOCHS | TIME (MIN) |
|---|---|---|
| CIRCLE | 2200 | 0.8 |
| PENDULUM | 500 | 3.7 |
| TEAPOT | 1000 | 8 |
| MIXTURE MODEL | 600 | 9 |
| TREE | 750 | 97 |
| FREY FACES | 1800 | 698 |
| DOUBLE PENDULUM | 1400 | 1801 |

Table 4: Training time for empirical tests. All networks were trained with 2617MB of RAM on a NVIDIA Titan X Pascal GPU.

## D METHODS COMPARISON

Competing methods were trained on the same dataset as for DyMoN, and with similar architecture where possible. We used Hidden Markov Models provided by the `hmmlearn` Python package and Kalman Filters provided by the `pykalman` package, in both cases learning all parameters by Expectation-Maximization (EM). The EM algorithm was run with default parameters. Recurrent Neural Networks were trained with the same number of hidden layers and convolutions as the DyMoN, except in the teapot example where we found performance improved by adding an additional two hidden layers of size 64. Performance of the competing methods using Earth Mover's Distance

|  | EMD | TRAINING (CPU S) | INFERENCE (CPU S / FORWARD PASS) |
|---|---|---|---|
| DYMON | $0.150 \pm 4\mathrm{e}{-4}$ | 9 MIN | 19.7 S |
| MCMC | $0.146 \pm 4\mathrm{e}{-4}$ | N/A | 26.2 S |
| RNN | $1.61 \pm 3.4\mathrm{e}{-3}$ | 269 MIN | 3209 S |
| KF | $0.357 \pm 1\mathrm{e}{-4}$ | 650 S | 15.1 S |
| HMM | $0.159 \pm 3\mathrm{e}{-4}$ | 92 S | 10.5 S |

Table 5: Performance of various learning methods for dynamical systems on generating a distribution of samples from the GMM dataset.

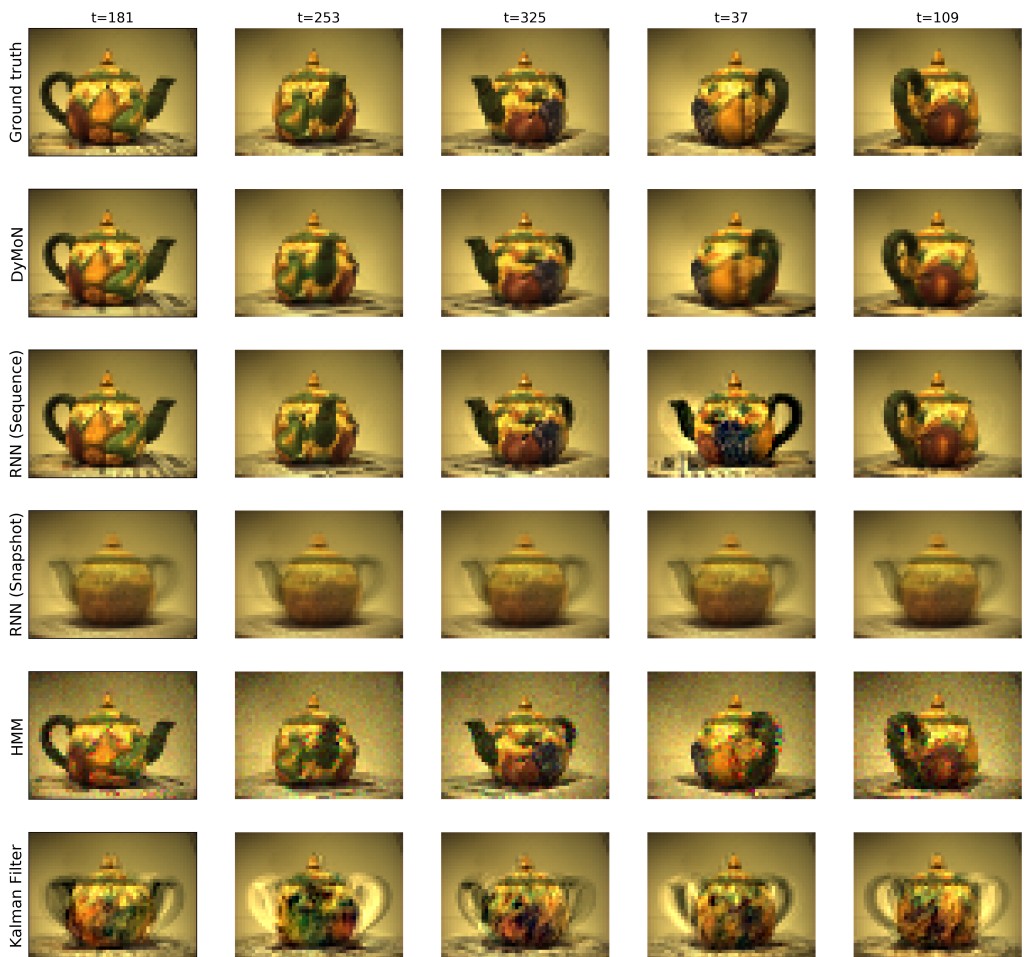

Figure 11: Examples of samples drawn from DyMoN, Recurrent Neural Network (RNN) given either a full sequence of inputs (Sequence) or a single input padded by zeroes (Snapshot), Hidden Markov Model (HMM) and Kalman Filter (KF) when trained on the teapot data.

(EMD) on the GMM example is shown in Table 5 and examples of generated frames from the teapot data are shown in Figure 11.

