# OpenReview forum: "Modeling Dynamics of Biological Systems with Deep Generative Neural Networks"
_ICLR.cc/2019/Conference_

### Official Review · AnonReviewer3 · 2018-11-02
**The paper details about a feed-forward NN, called DyMoN, used to model dynamic features of objects that are at multiple phases of transition. The authors need to relate how the Markovian processes are realized within the hidden units and why at all a deep architecture is required.**

**Rating:** 3
**Confidence:** 5

**Review:**


> Even though the paper details the underlying Markovian setup in Section 2, it is unclear to the reader how this knits with the FFNN architecture, for example what are the Markovian functions at hidden layer and output layer. Are they all conditional probabilities? How do you prove that this is what occurs within each node?

> Why is the functional form of f_\theta in Eq 1?

> How many hidden layers are in place?

> What is the Stochastic dynamical process in Figure A and how is this tethered to DyMon?

> The authors mention an nth-order Markovian process implemention but is this not the case with any fully connected neural network implementation? What the reader fails to see is why DyMoN is different to these already-existing architectures.

> In the teapot example, the authors mention a DyMoN architecture. (Page 8). Is this what is used throughout for all the experiments? If yes, why is it generalizable and if not, what is DyMoN’s architecture? You could open the DyMoN box in Figure 10 (1) and explain what DyMoN consists.


Section 2 is the crux of the paper and needs more work - explain the math in conjunction to the ‘deep’ architecture, what is the 'deep' architecture and why it is needed at all. Then go on to show/prove that the Markovian processes are indeed being realized.

---

### Official Review · AnonReviewer1 · 2018-11-04
**Novelty and performance evaluation are unclear**

**Rating:** 4
**Confidence:** 5

**Review:**

This paper describes a NN method called DyMoN for predicting transition vectors between states with a Markov process, and learning dynamics of stochastic systems.

Three biological case studies are presented, it is unclear if any new biology was learned from these cases that we could not have learned using other methods, and how accurate they are. The empirical validations are all on nonbiological data and disconnected from the first part of the paper, making the main application/advantage of this method confusing.

I agree with the computational advantages mentioned in the paper, however, interpretation of the representational aspect is challenging especially in the context of biological systems. Regarding denoising, what are the guarantees that this approach does now remove real biological heterogeneity? Also, a denoising method (MAGIC) was still used to preprocess the data prior to DyMon, there is no discussion about any contradictory assumptions.

Overall, the main shortcoming of the paper is lack of performance evaluation, comparison to other methods and clarifying advantages or novel results over other methods. The description of the method could also be improved and clarified with presenting an algorithm.

---

### Official Review · AnonReviewer2 · 2018-11-05
**Modelling dynamics of biological data with deep Neural Networks (NN)**

**Rating:** 6
**Confidence:** 2

**Review:**

This paper tackles the important challenge of making sense of temporal measurements made in biological systems. Among other, those have the peculiarity that they are not independent but with a dependency structure, which can be encoded as a graph or a network. The authors claim that their approach, DyMoN is adapted to the many challenges of biological high-throughput data sets: noise, sparsity and lack of temporal resolution. The paper presents three very different use of the method in complex biological systems in Section 3: (i) Calcium imaging of visual cortex neurons, (ii) T--cell development in the thymus, and (iii) Human embryonic stem cell differentiation. Section 4 assesses the performance of the method on simulated data sets as well as on a face-recognition data set. Moreover, the authors demonstrate how the features of the NN can be interrogated to shed new insight about the process under scrutiny. They also show the gain in running time a comapred to other approaches.

Remarks:
 - I know very little about the literautr in the subject, could you clarify how your work relates to/can be distinguished from: Testolin and Zorzi, Front Comput Neurosci. 201, Kiegeskorte's Ann. Rev. Vis. Sc. 2015 (https://doi.org/10.1146/annurev-vision-082114-035447), Kai Fan's PhD work (@ Duke University), Betzel and Bassel, Interface 2017, Wang et al. bioRxiv 2018 (https://doi.org/10.1101/247577), etc.
 - l-4p2: 'repetitions': what are those? Line below: 'sufficient for estimating $ P_{x}(y) $, means large sample size, no? No contradictory? And one line below: what is the precise meaning of 'similar' (twice)?
 - top p3: line continued from bottom of p2 -> is it to rubber out noise?
what is $ n $ in $ \mathbb{R}^{n} $?
 - Make Fig 1 (B) and (C) clearer, since the transition vectors are learnt, why are they in (B) (input states)?
 - below "...distribution approximates the distribution $ \mathcal{P}_{x} $..." -> but $ P_{x} $ also depends on $ \theta $, not an issue?
 - remark at the end of Section 2: extending DyMoN to higher-order: OK, but this might be computationally VERY expensive, don't you think?
 - link how your empirical validation data have features that, even remotely resemble those of the kind of biological data sets (on which no ground truth exist, I sympathise) you focus on.

---

### Meta-Review · Area_Chair1 · 2018-12-14
**Rejection, concerns not addressed by authors**

**Confidence:** 5
**Recommendation:** Reject

**Metareview:**

The paper tackles an interesting problem, which is effectively modeling biological time-series data. The advantages of deep neural networks over structured models like HMMs are their ability to learn features from the data, whereas probabilistic graphical models suffer from "model mismatch", where the available data must be carefully processed in order to fit the assumptions of the PGM. Any work advancing this topic would be extremely welcome in the world of machine learning in biology.

However, the reviewers each raised individual concerns about the paper regarding its clarity and quality, and the authors did not respond. Thus, the reviewers scores remain unchanged, and the rough consensus is a rejection.